# Air quality and related health impact in the UNECE region: source attribution and scenario analysis

Claudio A. Belis[1], Rita Van Dingenen[1]

[1] European Commission, Joint Research Centre, Via Fermi 2749, 21027 Ispra, Italy

*Correspondence to*: Claudio A. Belis (claudio.belis@ec.europa.eu)

**Abstract.** The TM5-FASST tool was used to study the influence of abatement policies within and outside the United Nations Economic Commission for Europe (UNECE) region on the exposure to $O_3$ and $PM_{2.5}$ and associated mortality in the UNECE countries. To that end, the impacts of pollutants deriving from different geographical areas and activity sectors were evaluated using ECLIPSE V6b air pollutant and greenhouse gases emission reduction scenarios. The mortalities were attributed to $O_3$

and $PM_{2.5}$ following the Global Burden of Disease approach and allocated to geographic areas (UNECE and non-UNECE) and activity sectors, including natural sources. In addition, a combination of runs designed for the purpose led to allocating exposure to $O_3$ and related mortality to two families of precursors: $NO_X$-VOC and $CH_4$. In this study the baseline scenario (CLE), which assumes that all air quality and greenhouse gas abatement measures adopted by 2018 are fully implemented, is compared with more ambitious scenarios (maximum feasible reduction, MFR). The findings from this comparison indicate

that $O_3$ exposure within the UNECE area is more sensitive to measures outside the UNECE region than $PM_{2.5}$ exposure, even though the latter leads to higher mortality than the former. In the "current legislation scenario" (CLE), the mortality associated with $O_3$ exposure in the UNECE region grows steadily from 2020 to 2050. The upward trend is mainly associated with the growing impact of $CH_4$ emissions from areas outside UNECE. Also, the mortality related to $NO_X$-VOC emissions outside UNECE increases in the same period. By comparison, a measurable decrease (13%) is observed in the mortality attributable

to NOx-VOC emissions within UNECE. In the same time window, the mortality associated with $PM_{2.5}$ exposure in the UNECE region decreases between 2020 and 2040 and then rises until 2050. The $PM_{2.5}$-related mortality in UNECE is mainly due to anthropogenic emissions within this region followed by natural sources (sea salt and dust) mainly located outside the UNECE region. Between 2020 and 2050, the impact of some UNECE anthropogenic sources on $PM_{2.5}$-related mortality decreases progressively, in particular road transport, energy production and domestic combustion while others, namely agriculture and

industry, show an upward trend. Finally, the analysis of MFR scenarios confirms that abatement measures in line with UN Sustainable Development Goals (SDGs) and the Paris Agreement can lead to significant co-benefits between air quality and climate policies.

## 1.    1 Introduction

In 2019, 6.67 million deaths globally (equivalent to 12% of the total deaths) were attributed to air pollution exposure, mainly due to fine particles and ozone (HEI, 2020). Air pollution is the main environmental risk of premature death worldwide.

However, the gap between low- and medium-income countries (LMIC) and high income countries (HIC) has widened since the beginning of this century due to the increasing trend of $PM_{2.5}$- related mortality in the former (Burnett and Cohen, 2020).

The Convention on Long-Range Transport of Air Pollution (also known as "the Air Convention") of the United Nations Economic Commission for Europe (UNECE) was adopted in 1979 and at present has 56 member States[1], including the EU since 1982. It has eight protocols, four of which are active. The Gothenburg Protocol to abate acidification, eutrophication and ground-level ozone is under review and an evaluation is in progress to assess the adequacy of its obligations and provisions. One of the aspects under evaluation is the future trend for improvements in air quality, human health and ecosystems impacts linked to methane ($CH_4$) emissions. Ground-level ozone ($O_3$) concentrations in most of the UNECE region countries are also influenced by other factors in addition to the regional ozone precursors: e.g. climatic parameters, hemispheric transport and global $CH_4$ emissions (Butler et al., 2020). Global background levels of $O_3$, $PM_{2.5}$ and their precursors, including $CH_4$ emissions, contribute significantly to air pollution within the UNECE region, with impacts on public health, ecosystems and biodiversity (Jonson et al., 2018; Lefohn et al., 2018). Projected trends in anthropogenic $CH_4$ emissions span a very wide range, depending on assumptions made about economic development and the use of emission control technology (Revell et al., 2015; Turnock et al., 2018).

The Air Convention protocols have contributed to reducing air pollution in UNECE countries. However, it is becoming more and more relevant to evaluate which pollutant levels are most affected/controlled by long-range transport of emissions outside the UNECE area, and to which extent new air quality guidelines can be achieved through emission reductions within UNECE only. This study aims to investigate to what extent the abatement policies within the UNECE region and in the rest of the world (ROW) influence the exposure to $O_3$ and $PM_{2.5}$ and associated mortality in the UNECE countries. To that end, the impacts of pollutants deriving from different geographical areas and activity sources that contribute to air quality related mortality in the UNECE region are analysed under different air pollutants and greenhouse gas (GHG) emissions' abatement scenarios. The emphasis is on quantifying the achievable benefits by analysing the gap between scenarios with different levels of ambition and the baseline. In particular, one of the scenarios (MFR BASE) is mainly driven by technological development connected to air pollutant emissions combined with a basic set of climate-oriented policies (national determined contributions) while the other scenario (MFR-SDS) is an archetype of the potentially achievable reductions by implementing the UN sustainable development goals related to energy combined with ambitious climate-oriented policies.

---

[1] Albania, Armenia, Austria, Azerbaijan, Belarus, Belgium, Bosnia and Herzegovina, Bulgaria, Canada, Croatia , Cyprus, Czech Republic , Denmark, Estonia, European Union, Finland, France, Georgia, Germany, Greece, Hungary, Iceland, Ireland, Israel, Italy, Kazakhstan, Kyrgyzstan, Latvia, Liechtenstein, Lithuania, Luxembourg, Malta, Monaco, Montenegro, Netherlands, North Macedonia, Norway, Poland, Portugal, Republic of Moldova, Romania, Russian Federation, Serbia , Slovakia , Slovenia , Spain, Sweden, Switzerland, Tajikistan, Türkiye, Turkmenistan, Ukraine, United Kingdom, United States of America and Uzbekistan.

## 2. Methods

### 2.1. Exposure and health impact assessment

The TM5-FAst Scenario Screening Tool (TM5-FASST) is a reduced-form air quality model based on linearised emission-concentration response sensitivities (also called source-receptor coefficients). The emission-concentration responses to regional emission changes were pre-computed at $1°x1°$ grid resolution with the full chemistry-transport (CTM) model TM5 (Krol et al., 2005) for 56 continental source regions, as well as for international shipping and aviation, for a 20% emission reduction in each of the relevant pollutant precursors ($SO_2$, $NOx$, $NH_3$, BC, OC, NMVOC) and for each individual source region. The resulting deviation relative to the unperturbed case in ground level pollutant concentrations is assumed to scale linearly with the emission deviation relative to the unperturbed case. More details are given in the S.I.

The TM5-FASST model bypasses CPU-expensive explicit chemical and physical process computations, at the cost of accuracy, as documented by Van Dingenen et al. (2018). It is worth mentioning that the model addresses impacts of anthropogenic emissions under constant meteorological conditions (year 2001), and therefore does not consider feedbacks of climate on photolysis rates, precursor residence times and deposition rates etc. This also implies that natural emissions of volatile organic components (including natural $CH_4$), $NOx$, as well as natural $PM_{2.5}$, are treated as fixed, constant contributions. Still, without claiming to be quantitatively equivalent to a full CTM, the model captures major features and implications of emission trends and has proven to be a useful screening tool in science-policy analysis (Van Dingenen et al., 2018).

A great advantage of a source-receptor model is that it keeps track of the contribution of each of the 56+2 source regions, as well as each individual precursor, to each receptor grid cell of the global domain, under the first-order assumption that all contributions can be added up linearly. This makes the model particularly useful for source attribution studies, which can be applied with a large flexibility on the definition of the receptor regions, the latter being a customisable aggregation of grid cells. In this study we consider as receptor region the UNECE domain, and we explore contributions of pollutant emissions outside and inside the UNECE region. Further detail in the attribution studies is obtained by breaking down the emissions by anthropogenic sector.

Health-relevant exposure metrics considered in the present study are the population-weighted $PM_{2.5}$ concentrations (as the sum of sulphate, nitrate, ammonium and primary $PM_{2.5}$) and the seasonal daily maximum 8h ozone average (SDMA8h). We apply a sub-grid correction to account for the spatial correlation between population density and primary $PM_{2.5}$ associated with transport and household emissions, leading to a higher estimated exposure than the value based on a uniform $PM_{2.5}$ distribution across the $1°x1°$ grid. This is relevant where strong population gradients occur within a single grid (Van Dingenen et al., 2018). Details of the applied parametrisation are given in the S.I.

The mortality associated with exposure to outdoor pollutants is estimated according to the Global Burden of Disease (GBD) approach (Stanaway et al., 2018). The methodology to estimate air quality-health impacts is given in the S.I. A complete description and validation of the TM5-FASST model is provided in Van Dingenen et al. (2018) and Belis et al. (2022).

## 2.2.        Sources

The contributions (or impacts) from anthropogenic sectors and natural emissions to PM$_{2.5}$ and O$_3$ exposure metrics in the UNECE region are estimated by the so called brute-force or emission reduction impact approach (Belis et al., 2020; Belis et al., 2021). The impact of the following anthropogenic activity sectors (11) was quantified: agriculture (AGR), agricultural waste burning (AWB), domestic and commercial combustion (DOM), energy production (ENE), industry (IND), use of

95   solvents (SLV), road transport (TRA), gas flaring (FLR), waste management (WST), open biomass burning (BMB) and maritime (SHP). Historical fire emissions were added from van Marle et al. (2017) and projections from the harmonized CMIP6 SSP2 scenario (Feng et all., 2019), including large-scale biomass burning and savannah burning and excluding AWB emissions to avoid double counting with the ECLIPSE V6b emissions. The resulting anthropogenic PM$_{2.5}$ concentration fields are overlaid with fixed natural PM$_{2.5}$ sources dust (DUST) and sea salt (SS), taken as the average of the CAMS reanalysis for

100   years 2000 to 2008 (https://www.ecmwf.int/en/forecasts/dataset/cams-global-reanalysis). For O$_3$, the abovementioned sectoral attribution was complemented with runs separating the impact of NO$_X$–NMVOC (hereon NO$_X$–VOC) and CH$_4$ precursor emissions from: (1) UNECE (continental, anthropogenic), (2) ROW (rest of the world: non-UNECE continental, anthropogenic), (3) international shipping (hereon maritime), and (4) other sources, according to the scheme described in Figure 1 .

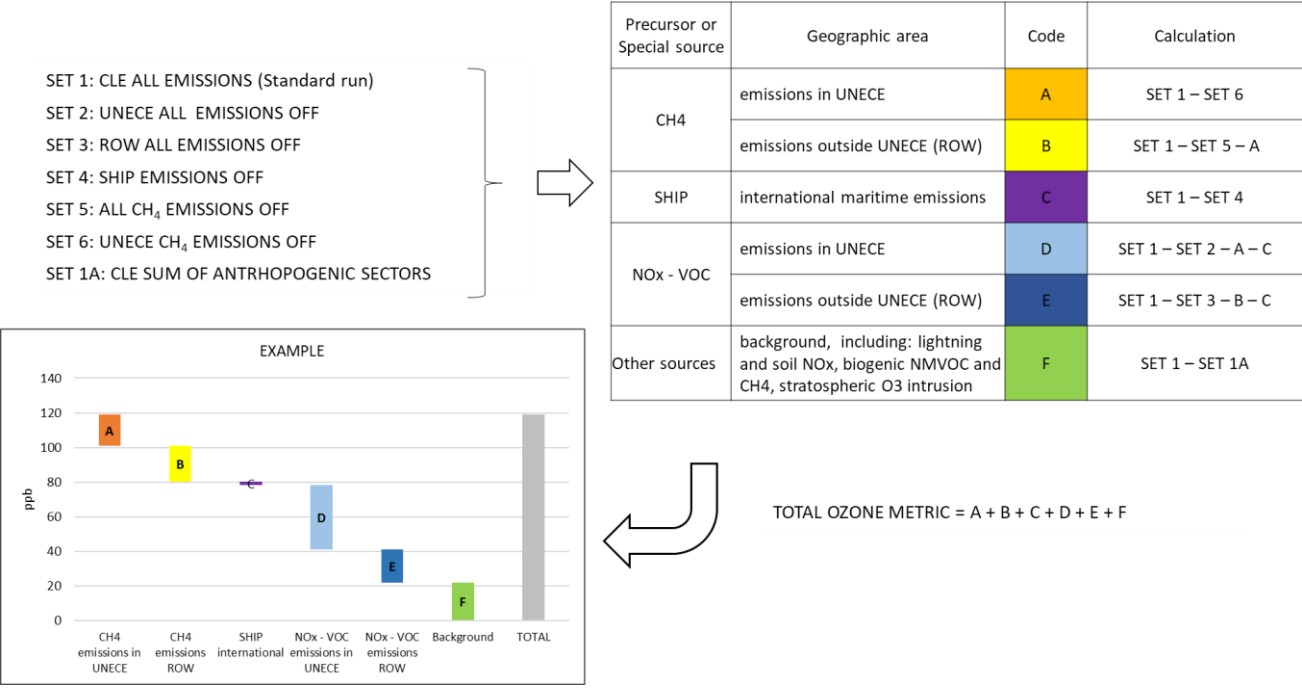

105

**Figure 1. Approach adopted to split O$_3$ concentrations by emission area (UNECE and non UNECE (ROW)) and by precursor (NO$_X$–VOC and CH$_4$).**

The standard simulations (Set 1) include the emissions in each of the three ECLIPSE V6b scenarios as described in section 2.3 (CLE, MFR BASE, MFR-SDS). In addition, a series of perturbations (sets 2 to 6) were computed in which the emissions of specific $O_3$ precursors (either $NO_X$–VOC or $CH_4$) were reduced worldwide or in specific areas (either UNECE or rest of the world) for each of the abovementioned scenarios. A total of 18 simulations were computed: one for each of the three scenarios in each of the six sets. Subsequently, sets 1 to 6 were conveniently subtracted as described in Figure 1 (right) to split the contributions/impacts of UNECE countries from those of the rest of the world and allocate $O_3$ to its two families of precursors: $NO_X$–VOC or $CH_4$. International maritime emissions are allocated into a stand-alone category as they are not attributed to any geographic area (UNECE nor ROW). The category "other sources" includes emissions not allocated to any specific area nor precursor (e.g. lightning and soil $NO_X$, biogenic NMVOC and $CH_4$, stratospheric $O_3$ intrusion). In the analysis of the results, the apportionment by region and precursor described here was combined with the information about anthropogenic sources described at the beginning of this section.

In Appendix A, the $PM_{2.5}$ and $O_3$ source apportionment presented in this study is compared with similar studies in the literature. The obtained shares for the $PM_{2.5}$ and $O_3$ exposure metrics are converted to total mortalities according to:

$$MORTALITY_{source\ x} = MORTALITY_{total} \times \frac{EXPOSURE\ METRIC_{source\ x}}{EXPOSURE\ METRIC_{total}} \qquad (4)$$

Where *EXPOSURE METRIC $_{total}$* is the sum of all individual sources (*x*) shares.

## 2.3.    Scenarios

This study evaluates a set of scenarios (Appendix A, Table A1) derived from the ECLIPSE dataset version 6b (Amman et al., 2011; Klimont et al., 2017) developed using the GAINS model (IIASA, 2022). To assess different levels of ambition in the abatement policies from 2020 onwards the CLE is compared with two maximum feasible reduction (MFR) scenarios: MFR BASE and MFR-SDS (Appendix A, Table A1). For every macro-sector (e.g. energy, transport, industry, etc.), each scenario combines a set of cross-cutting measures with others specific for each region of the world. The CLE and MFR BASE scenarios are based on the International Energy Agency (IEA) new policy scenario (NPS; IEA, 2018) which includes measures that had been announced by 2018 and makes no assumptions about further evolution of these positions nor aims to achieving any particular outcome. The NPS includes European Union's 2030 renewable energy and energy efficiency targets, the Chinese three-year action plan for cleaner air, the planned revision of the Corporate Average Fuel Economy standards in the United States, as well as the announced US Affordable Clean Energy rule. Moreover, it considers Japan's revised basic energy plan and Korea's 8th National Electricity Plan. The climate policy for both CLE and MFR BASE is the same and is specific for every country as it is based on the countries' national determined contributions (NDCs) under the Paris agreement (https://unfccc.int/process-and-meetings/the-paris-agreement/nationally-determined-contributions-ndcs). Example of cross-cutting measures in the NPS are: fuel sulphur standards of 10-15 ppm in the road transport sector, global cap of 0.5% on

sulphur content in fuel in 2020 in the international shipping sector, and improving fuel efficiency by 2% per year until 2020 in the international aviation sector. The emission reduction in the MFR BASE scenario compared to the CLE are based on the introduction of best available technology with no cost limitations (Table A1).

Unlike the previous two, the MFR-SDS scenario is based on the IEA's Sustainable Development Scenario (IEA, 2018) which includes the main energy-related components of the Sustainable Development Goals, agreed by 193 countries in 2015 to keep the increase of global average temperature below 2 °C, achieving universal access to modern energy by 2030 and reducing dramatically the premature deaths due to energy-related air pollution. Examples of cross-cutting assumptions in the SDS are: staggered introduction of $CO_2$ prices, fossil fuel subsidies phased out by 2025 in net-importing countries and by 2035 in net-exporting countries, and maximum sulphur content of oil products capped at 1% for heavy fuel oil, 0.1% for gasoil and 10 ppm for gasoline and diesel. A full description of the scenarios goes beyond the purposes of the present work. More details are available elsewhere (https://iiasa.ac.at/web/home/research/researchPrograms/air/ECLIPSEv6.html; IEA, 2018; Belis et al., 2022).

In this study were used the SSP gridded population projections from Jones and O'Neill (2016). The SSP2 projections were associated with CLE and MFR BASE while SSP1 were used with the MFR-SDS scenario.

## 3.    Results

### 3.1.    Emissions

The UNECE and ROW emission trends between 2020 and 2050 of $O_3$ and $PM_{2.5}$ precursors in all the studied scenarios are shown in Figure 2. In the **CLE scenario**, UNECE $NO_X$, NMVOC and $PM_{2.5}$ emissions decrease by 33%, 13% and 13%, respectively, between 2020 and 2050 while in ROW, $NH_3$ and $CH_4$ grow by 27% and 34%, respectively.

In both MFR scenarios, UNECE emissions show a downward trend over the whole time window with the exception of $NH_3$, which after an initial decrease remains stable. Moreover, in these scenarios $NH_3$ is the only precursor with a distinguishable upward emission trend between 2025 and 2050 in ROW while all the others show a downward trend. In MFR BASE, UNECE emissions in 2050 are between 69% ($PM_{2.5}$) and 35% ($NH_3$) lower than CLE while ROW emissions are between 80% ($PM_{2.5}$) and 37% ($NH_3$) lower than CLE. Despite MFR-SDS emissions follow similar trends, the reductions with respect to the CLE are higher, with the exception of $NH_3$ which is the same in both MFR scenarios.

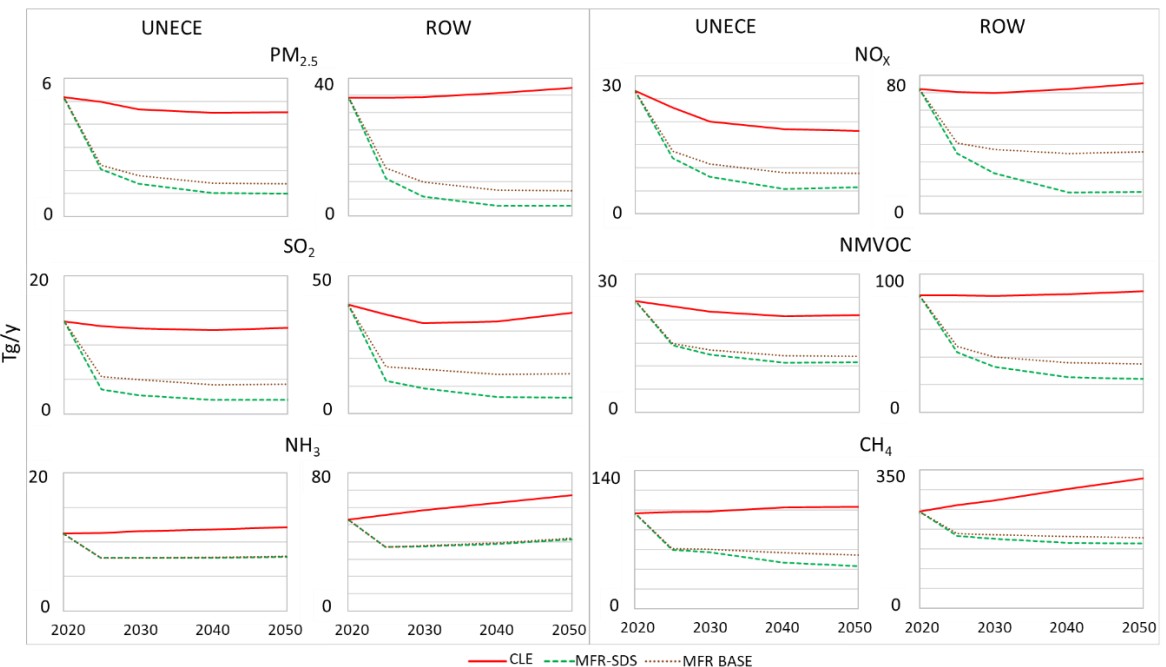

**Figure 2. UNECE (left) and ROW (right) emission trends of main O₃ and PM₂.₅ precursors in the studied ECLIPSE V6b scenarios**

### 3.2. Influence of ROW on UNECE

To assess the impact of air pollutant and GHG abatement measures outside the UNECE region (Rest of the World; ROW) on
170    UNECE emission abatement policies, a regional source attribution exercise is discussed in this section. The exposure to PM₂.₅ (anthropogenic) and O₃ in UNECE countries between 2020 and 2050 in the global baseline scenario (CLE) is compared with the MFR BASE scenario and with a scenario in which the emission reductions foreseen in the MFR BASE are applied only in the UNECE region while CLE emissions are kept in ROW (MFR UNECE scenario) (Figure 3).

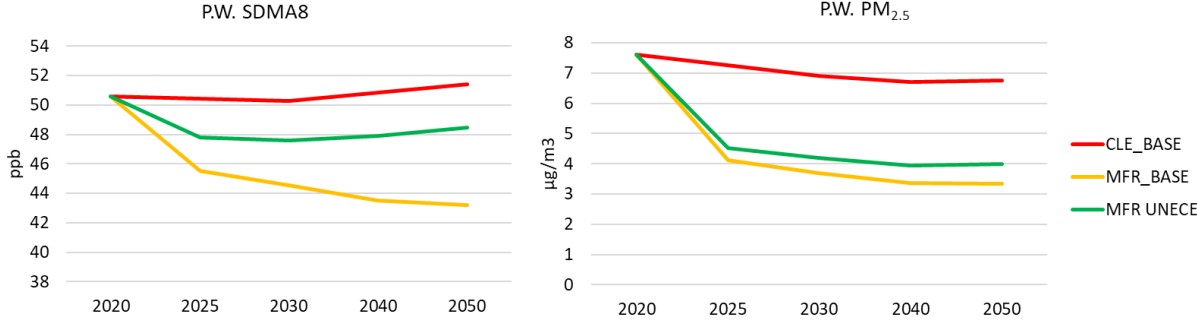

175    **Figure 3. O₃ seasonal mean of 8hr (population weighted SDMA8h, left) and anthropogenic population weighted PM₂.₅ (right) annual averages in UNECE region, average of countries, under different scenarios. CLE (current legislation), MFR BASE applied in UNECE countries only (MFR UNECE), MFR BASE in all countries (MFR BASE).**

The $O_3$ exposure in CLE (red line) and MFR UNECE (green line) shows an upward trend from 2025 onwards. The abatement benefit, i.e. the difference between the $O_3$ exposure in CLE and MFR UNECE, over the considered time window is relatively small (5% to 6%) suggesting that applying emission reductions in UNECE countries only, leads to limited additional abatement in the $O_3$ exposure in UNECE countries relative to the baseline (CLE). By comparison, the $O_3$ exposure in MFR BASE (yellow line) follows a downward trend and the abatement benefit (delta CLE-MFR BASE) is twice as much as MFR UNECE (10% to 16%) indicating that implementing MFR worldwide would not only lead to higher abatement of exposure in UNECE but also reverses the trend from increasing to decreasing.

Unlike $O_3$, $PM_{2.5}$ exposure shows a decreasing trend for the three scenarios. The abatement benefit (CLE - MFR UNECE) over the studied period is already high (-38% to -41%) and applying the MFR BASE scenario globally leads to a relatively small marginal benefit ($\leq$ 10% of CLE). In synthesis, for $PM_{2.5}$ abatement, UNECE is only slightly affected by ROW measures, while $O_3$ levels are strongly modulated by measures taken outside the UNECE region. This is obviously related to the longer (compared to $PM_{2.5}$) atmospheric lifetime of $O_3$, formed from its short-lived precursors $NO_x$ and NMVOC, and of its long-lived precursor $CH_4$ which contributes to global background $O_3$. The UNECE countries where the differences in $O_3$ and $PM_{2.5}$ exposure between MFR UNECE and MFR BASE are the highest (in the range 6 to 10 ppb and 1.5 to 2.4 µg/m$^3$, respectively) are located at the boundary of the UNECE region and therefore more exposed to long-range pollution from the ROW (Figure S2). Some of these countries are in the Caucasus and central Asia (Armenia, Azerbaijan, Tajikistan, Kyrgyzstan and Turkmenistan) downwind highly polluted regions (e.g. southern Asia, Far East). The highest differences between these scenarios for both pollutants are observed in Israel which is a small country surrounded by an area of non-UNECE countries with high emissions. Some countries in the Atlantic coastal area (Portugal, Spain and Ireland) present high differences in the $O_3$ exposure between MFR UNECE and MFR BASE likely due to the influence of air masses circulating over the sea and mostly affected by emissions in ROW. A similar situation is observed in Malta which is mostly affected by the high background levels in the Mediterranean Sea.

The attribution of $O_3$ and $PM_{2.5}$ levels to precursor emissions in- and outside the UNECE region is further investigated in the following sections.

### 3.3.    Source allocation of ozone exposure and premature mortality in UNECE in the baseline scenario (CLE)

In this section, the $O_3$ exposure and related mortality within UNECE is broken down by (a) precursor (b) sector and (c) source region (UNECE vs. ROW) considering only the attribution runs of the CLE scenario. The $O_3$ background (OTHER/NATURAL), including biogenic and other unspecified sources (Figure 4a), is estimated by subtracting the sum of all anthropogenic sectors from total $O_3$ (see section 2.2) and is the main single contributor to the $O_3$ exposure. The impact of this "source" is approximately 35 ppb and remains relatively constant throughout the analysed time window (2020 – 2050). Despite its dominance, this component is not the main focus of the analysis since it is, by design, little affected by anthropogenic emissions in the short term. In the 2020 – 2050 time window, the anthropogenic fraction of the $O_3$ exposure is worth 16 - 19 ppb.

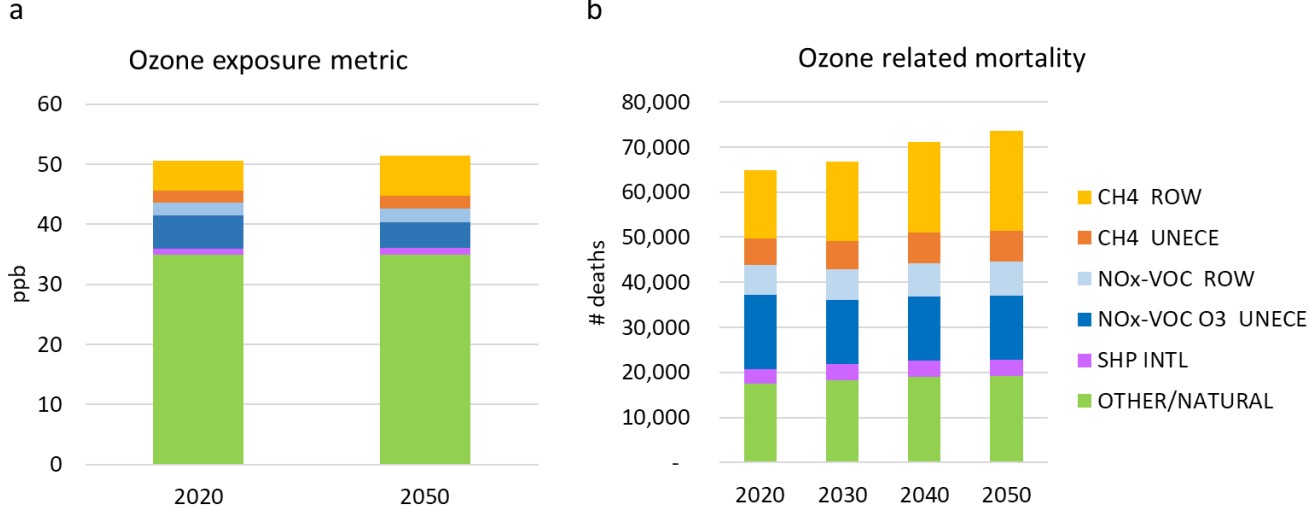

**Figure 4. Allocation of the population weighted O₃ (SDMA8h) exposure in UNECE to geographic source areas (UNECE, ROW), precursors and other/natural sources. Units: ppb (a). Mortality (UNECE total) associated with O₃ exposure in UNECE split by natural-background (only the fraction above the exposure threshold) and anthropogenic emissions (b).**

In terms of precursors, in CLE there is a remarkable shift in the relative role of short-lived components ($NO_X$, NMVOC) versus $CH_4$ between 2020 and 2050. The initial dominant role of $NO_X$ and NMVOC in anthropogenic ozone formation is replaced by $CH_4$ towards 2050. This is due to the combined decrease of UNECE $NO_X$ and NMVOC emissions (while ROW emissions remain relatively constant) and the increase of ROW $CH_4$ emissions (while UNECE emissions remain relatively constant). The overall O₃ exposure metric is stable along the observed time window because the decreasing impact of $NO_X$-VOC

emissions from UNECE over time is largely compensated by the increasing impact of $CH_4$ emitted in ROW.

The overall share of O₃ exposure allocated to anthropogenic $NO_X$-VOC emissions is mainly associated with transport, industry and maritime sources while the $CH_4$ emissions affecting this pollutant are mainly emitted from agriculture, gas flaring and waste management. Energy production, another important anthropogenic source, presents similar shares of both precursor families (Figure S3).

In Figure 4b the premature mortality associated with O₃ exposure in the UNECE region estimated in the CLE is shown. The number of premature deaths grows steadily from 65,000 in 2020 to 74,000 in 2050. This upward trend in mortality is mainly associated with an increased impact of anthropogenic $CH_4$ emissions from ROW (+46 %, +7,000 deaths/year). Also the mortality related to anthropogenic $NO_X$-VOC emissions in ROW increases by 17% in the same period (+1,000 deaths/year). On the contrary, a measurable decrease is observed in the mortality attributable to anthropogenic $NO_X$-VOC emissions in

UNECE which drops from 16,000 in 2020 to 14,000 in 2050.

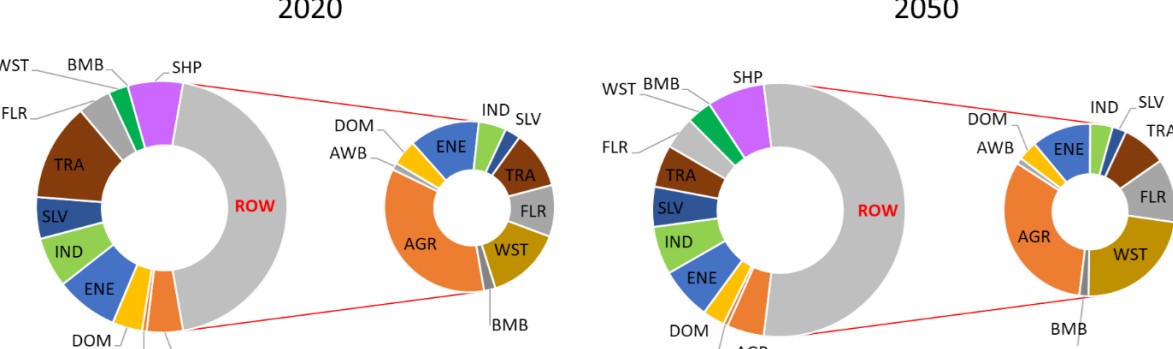

**Figure 5. Allocation of O₃ exposure and related mortality (UNECE avg.) to anthropogenic sources under CLE. The overall impacts are represented in the main pie charts while the small pie charts to the left of them show the detail of ROW impacts only. The data are also available in Table S1. AGR: agriculture, AWB: agricultural waste burning, DOM: domestic and commercial combustion, ENE: energy production, IND: industry, SLV: use of solvents, TRA: road transport, FLR: gas flaring, WST: waste management, BMB: open biomass burning and SHP: maritime.**

The contributing sectors change their relative importance evolving from a mix dominated by transport, agriculture and energy production in 2020 to a one dominated by agriculture, waste management, transport and energy production in 2050 (Figure 5, Table S1). Transport, industry and maritime contribute to O₃ exposure only via NO$_X$-VOC precursors while agriculture, gas flaring and waste management contribute almost only via CH₄ emissions (Figure S3).

The CH₄ impact of agriculture, gas flaring, waste management and energy production emissions from ROW on O₃ exposure in UNECE presents an upward trend between 2020 and 2050 (Figure S3). In the same time window, the NO$_X$-VOC contribution from transport, energy production and domestic emissions from UNECE show a downward trend with the exception of industry which increases slightly. Although energy production is the only source which shares of O₃ exposure due to NO$_X$-VOC and CH₄ are comparable, the balance between these two components evolves along the studied time window towards an increase in the share of the latter.

## 3.4.    Source allocation of PM₂.₅ exposure and premature mortality in UNECE in the baseline scenario

The UNECE anthropogenic emissions are the main responsible for PM₂.₅ exposure in UNECE, with a decreasing trend between 2020 and 2050, while those from ROW have a minor role which increases slightly over the observed time window (Figure 6).

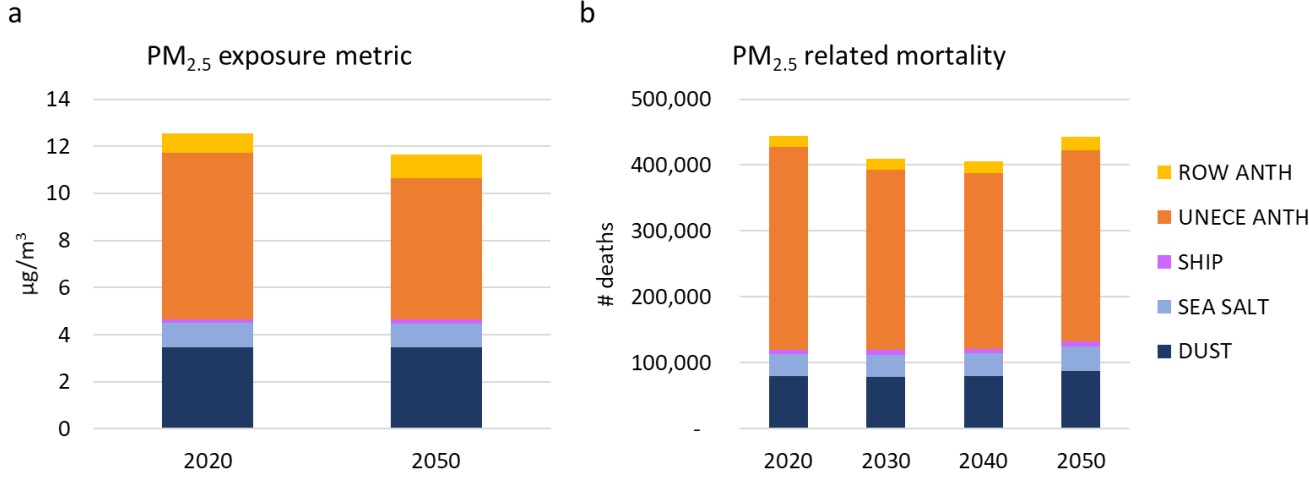

a

## PM$_{2.5}$ exposure metric

b

## PM$_{2.5}$ related mortality

Legend: ROW ANTH, UNECE ANTH, SHIP, SEA SALT, DUST


**Figure 6. Allocation of the population weighted PM$_{2.5}$ exposure in UNECE to geographic source areas (UNECE, ROW) and natural sources under CLE (a). Mortality (UNECE avg.) associated with PM$_{2.5}$ exposure attributable to both anthropogenic and natural sources under CLE (b).**

The mortality associated with PM$_{2.5}$ exposure in the UNECE region (including both natural and anthropogenic sources) is

444,000 cases in 2020. It shows a downward trend between 2020 and 2030 and a subsequent rise between 2040 and 2050 when it reaches 443,000 units (**Error! Reference source not found.**Figure 6b).

The main anthropogenic contributors within UNECE are: agriculture, industry, domestic, energy production and transport (Figure 7, Table S2). An overall downward trend in the impact of domestic, energy production and transport from UNECE and an increasing role of industry and agriculture from this region are observed. The share of maritime, a contributor which is

not geographically allocated in this analysis, is stable from 2020 onwards. In 2050, there is an increase in the PM$_{2.5}$ exposure mainly due to a rise in the impact of agriculture, transport, gas flaring and waste management emissions from ROW and agriculture and industry emissions from the UNECE region.

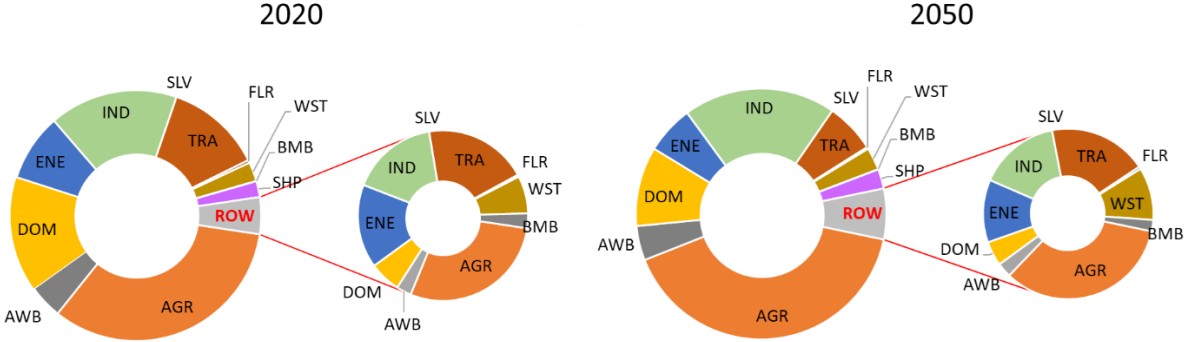

2020 | 2050

**Figure 7. Allocation of PM$_{2.5}$ exposure and related mortality (UNECE avg.) to anthropogenic sources under CLE. The overall**
**impacts are represented in the main pie charts while the small pie charts to the left of them show the detail of ROW impacts only.**

The data are also available in Table S2. AGR: agriculture, AWB: agricultural waste burning, DOM: domestic and commercial combustion, ENE: energy production, IND: industry, SLV: use of solvents, TRA: road transport, FLR: gas flaring, WST: waste management, BMB: open biomass burning and SHP: maritime.

### 3.5.    Source allocation of exposure to air pollutants in UNECE in MFR scenarios

This section evaluates the trends of the $O_3$ and $PM_{2.5}$ exposure in UNECE between 2020 and 2050 computed with TM5-FASST using the ECLIPSE V6b MFR BASE and MFR-SDS emission scenarios (Table A1; Figure 8). In 2050, the MFR BASE and MFR-SDS $O_3$ exposure is 16% and 20% lower than CLE, respectively, while the $PM_{2.5}$ (anthropogenic) exposure in the abovementioned scenarios is 51 % and 59% below CLE, respectively.

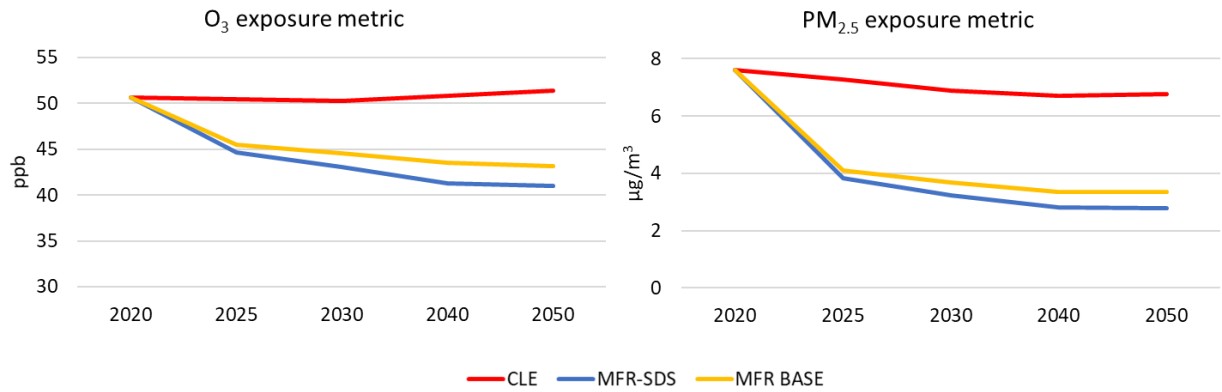

**Figure 8. $O_3$ and anthropogenic $PM_{2.5}$ exposure metrics (UNECE avg.) computed with TM5-FASST according to the ECLIPSE V 6b scenarios: CLE, MFR BASE and MFR-SDS.**

In the period 2025 – 2050, the main anthropogenic contributor to $O_3$ exposure and mortality in both MFR scenarios is by far agriculture due to $CH_4$ emissions in ROW (Figure S4).

In the MFR BASE scenario, which is mainly based on the implementation of best available technologies (BATs) and Paris

Agreement NDCs, the delta mortality in UNECE compared to CLE ranges from -13,000 cases (-21%) in 2025 to -24,000 cases (-34%) in 2050 due to lower $O_3$ exposure (Figure 9 top left). Such improvement is mainly associated with $NO_X$-VOC emission reductions in the UNECE region and reductions of $CH_4$ in ROW, the role of which increases considerably between 2025 and 2050 (Figure 9 top left). A more detailed analysis of the MFR BASE reveals that the main UNECE $NO_X$-VOC emission reductions in 2050 are associated with energy production, industry and transport sectors. By comparison, those of $CH_4$ in ROW

are mainly due to abatement of gas flaring and energy production in 2025 with dramatic abatement increase in the waste management sector between this year and 2050 (Figure 10 top).

The additional improvement compared to the MFR BASE from the most ambitious MFR-SDS scenario, in line with energy related SDGs and global temperature increase containment, ranges between ca. -2,000 cases (-4%) in 2025 and -5,500 (-11%) cases in 2050 and is mainly due to the reduction of $NO_X$-VOC emissions in both UNECE and ROW (Figure 9 bottom left).

Such abatement of $O_3$-related mortality in the MFR SDS scenario is associated with emission reductions in the transport sector in 2050 in both UNECE and ROW compared to 2020 (Figure 10 bottom).

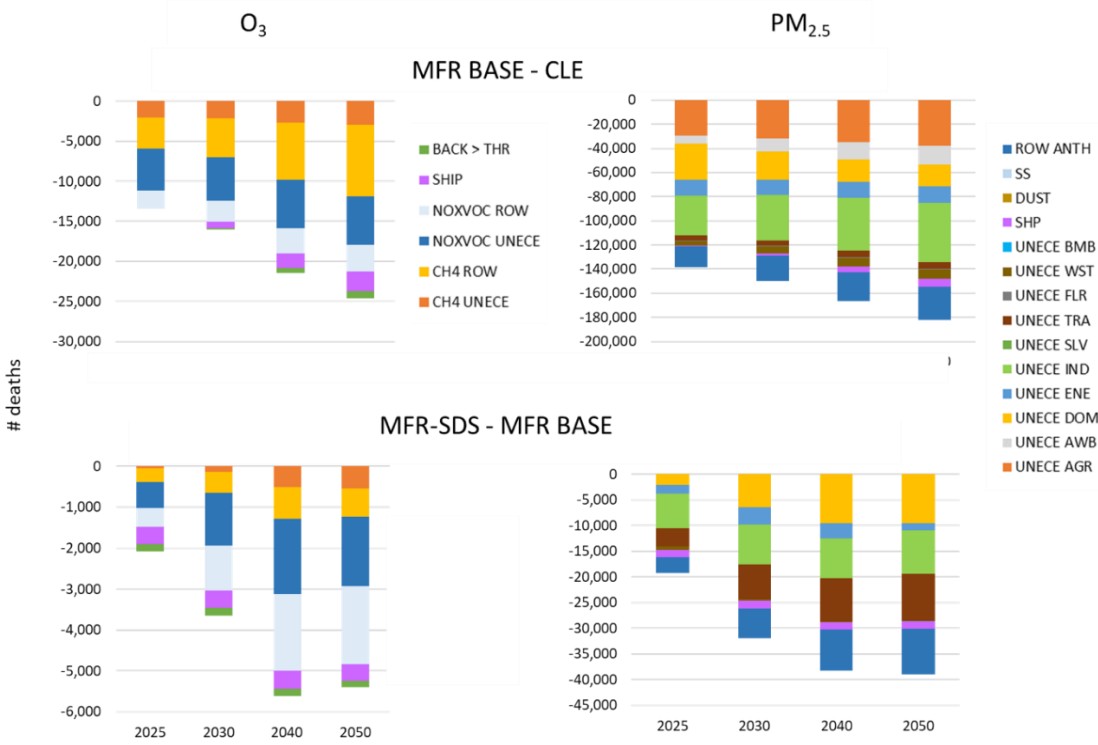

**Figure 9. Delta MFR BASE - CLE and MFR-SDS – MFR BASE of O₃ (left) and PM₂.₅ (right) associated mortality (UNECE total) split by precursor and main emission areas. For O₃ we only consider the fraction of 'OTHER/NATURAL' exceeding the zero effect threshold of 29.1 ppb.**

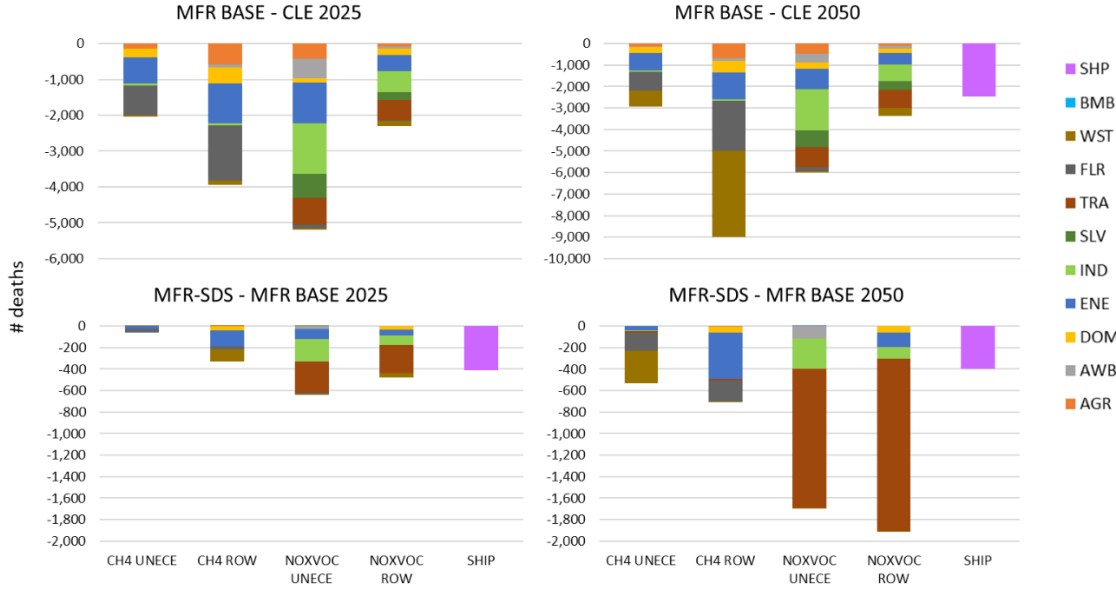

**Figure 10. Delta MFR BASE – CLE (top) and MFR-SDS – MFR BASE (bottom) of UNECE O₃ associated mortality in 2025 and 2050 split by source sectors.**

In MFR BASE the delta mortality in UNECE due to $PM_{2.5}$ exposure compared to the CLE ranges from ca. -137,000 cases (-33%) in 2025 to ca. -187,000 cases (-41%) in 2050 (Figure 9 top right). Such improvement is mainly due to abatement of emissions in the agriculture and industry sectors in UNECE. In this region, the abatement of emissions in the domestic sector shows a decreasing importance between 2025 and 2050 while the opposite is true for agricultural waste burning and the anthropogenic emissions in ROW. By comparison, the MFR-SDS scenario leads to an additional reduction in mortality compared to the MFR BASE of ca. -19,000 cases (-7%) in 2025 that reaches ca. -40,000 cases (-15%) in 2050 (Figure 9 bottom right). In this case, the reduction is associated with industry emissions abatement, relatively constant throughout the observed period, and an increasing abatement along the studied time window in domestic and transport sectors from UNECE and anthropogenic emissions in ROW (Figure 9 bottom right).

## 4.       Main findings and discussion

Implementing more stringent air quality and GHG emission abatement policies only in the UNECE region (MFR UNECE scenario) leads to limited benefits in the air pollution exposure in this region because their effect is partially offset by the unabated emissions from non-UNECE countries, when similar measures are not implemented there as well. Such effect is more pronounced for $O_3$ than for $PM_{2.5}$.

In CLE, the main single contributor to the $O_3$ exposure in the UNECE region is non-anthropogenic $O_3$ (OTHER/NATURAL), including biogenic and other unspecified sources (mainly soil-derived $NO_X$, lightning and stratospheric intrusion), which remains relatively constant at ca. 35 ppb throughout the entire time window (2020 – 2050). In this scenario, the anthropogenic fraction of the $O_3$ exposure is equivalent to 16 - 19 ppb. Transport, industry and maritime sectors contribute to this fraction mainly via $NO_X$-VOC precursors' emissions while agriculture, gas flaring and waste management contribute mostly via emissions of the $CH_4$ precursor. Energy production is the only source affecting $O_3$ exposure with similar shares for both precursor families.

The overall upward trend in the $O_3$ related mortality in the UNECE region over the studied time window is mainly associated with the increasing share of $CH_4$ emissions from ROW. The $O_3$ exposure shares of agriculture, waste management , gas flaring and energy production $CH_4$ emissions from ROW shows an upward trend along the simulated time window while the one of transport, energy production and domestic $NO_X$-VOC emissions from UNECE shows an opposite trend.

Unlike $O_3$, anthropogenic UNECE emissions are the main source of $PM_{2.5}$ exposure and related mortality in UNECE countries. However, due to a reduction in the share of UNECE emissions and an increase in that from ROW, the importance of the former decreases from 70% to 65% of the total $PM_{2.5}$ exposure metric over the simulated time window.

As a whole, the MFR BASE leads to 34% and 41 % mortality reductions compared to the CLE scenario in 2050 for $O_3$ and $PM_{2.5}$ exposure, respectively, while the MFR-SDS leads to a total abatement of mortality in 2050 compared to CLE of 41% and 50% for $O_3$ and $PM_{2.5}$ exposure, respectively.

The applied methodology, based on a reduced form model, has several limitations we discuss here. Some of the limitations are inherited from the parent TM5 CTM. This is the case for secondary organic aerosol chemistry which is not considered and leads to a conservative estimate of $PM_{2.5}$ exposure and consequently of the benefits from controls. The omission of secondary organic PM in TM5 is estimated to introduce a low bias in the $PM_{2.5}$ concentration of the order of 0.1 μg/m$^3$ as global mean. However, regional levels in central Europe and China can reach up to 1 μg/m$^3$ in areas where average levels of primary organic

matter are 20 μg/m$^3$ (Van Dingenen et al., 2018). In addition, the TM5-FASST model does not include non-linear responses due to changing chemical regimes when switching off individual precursor emissions, nor does it consider impacts of future climate change on photolysis rates and on natural emissions that may affect ozone chemistry. Although an evaluation of climate-chemistry interactions is beyond the capabilities and the scope of the TM5-FASST model, we briefly discuss their possible impacts on our conclusions. The interaction between pollution and climate is complex as it involves many processes:

meteorology, precursor emissions and atmospheric chemistry, the interactions of which introduce a considerable degree of uncertainty. For instance, the $O_3$ level derives from the emission of precursors ($NO_x$ and VOC including methane), the interplay of which determines the chemical regime ($NO_x$-limited or VOC-limited), and is modulated by temperature. From a purely meteorological point of view, a warmer climate is expected to cause a higher frequency of stagnant conditions leading to higher surface ozone production due to higher photolysis rates which would call for more stringent controls than anticipated under

present climate in order to meet limit levels. Such climate penalty on summertime surface ozone concentrations is estimated to be in the range 1 – 10 ppb, with highest impacts in polluted conditions (Jacob and Winner, 2008). However, precursor emission reductions may decouple the long-term trends of $O_3$ summer maximum concentrations and temperature (Fiore et al., 2015).

Natural VOC emissions from vegetation are expected to increase with increasing temperature – up to a critical level after

which emissions decrease again (e.g. 38°C for isoprene). Moreover, the VOC emissions are species-specific and therefore subject to changes due to type of vegetation or land use variations (Wu et al., 2012). In $NO_x$-saturated (VOC-limited) conditions (typical of urban polluted areas), the climate-driven increased VOC emissions would increase the natural component of $O_3$ formation, and drive the chemical regime more towards the $NO_x$-limited region, implying a higher response of $O_3$ to anthropogenic $NO_x$ emission changes. However, under the more common conditions of VOC-saturation ($NO_x$-limitation), the

$O_3$ response to $NO_x$ is only weakly dependent on the VOC concentrations (Akimoto and Tanimoto, 2022).

A warming climate is also expected to increase $CH_4$ emissions from wetlands, the major natural $CH_4$ source (Gedney et al., 2004), however, the magnitude of such variation is still quite uncertain (Nisbet, 2023).

The applied TM5-FASST methodology, not including these climate-chemistry feedbacks, is likely to underestimate the natural component of $O_3$ formation in a future, warmer climate, as well as the $O_3$ response to $NO_x$ reductions in specific polluted

conditions. However, this does not compromise our conclusion that control of anthropogenic $CH_4$ emissions can play a prominent and increasing role in the coming decades.

The estimated levels and source allocation in our study are comparable with those obtained in studies with similar scope. However, using previous studies as reference is not straightforward due to different underlying methodological assumptions

and aggregation of the output data. This is particularly true when comparing the source apportionment with brute-force or emission reduction impact approach (used in this study) with the one resulting from tagged method studies (Appendix A, Figure A2).

## 5.    Conclusions

The scenario analysis presented in this study assesses the exposure to $O_3$ and $PM_{2.5}$ and associated mortality between 2020 and 2050 in the UNECE countries. To that end, a baseline scenario in which the air quality and GHG abatement measures adopted by 2018 are implemented (CLE) is compared with other scenarios with increasing degree of ambition. The adopted methodology for the identification of geographical origin with sectoral anthropogenic sources and precursors detail led to an in-depth understanding of the impact that different measures may have on mortality in the UNECE region in the medium and long-term.

The study demonstrates that applying emission reductions only in UNECE countries leads to a limited abatement in the $O_3$ exposure in UNECE countries with respect to the baseline (CLE) and that the implementation of BATs worldwide would not only lead to higher abatement of exposure in UNECE countries but also to a trend reversal, from increasing to decreasing. Moreover, the study shows that the overall upward trend in the $O_3$-related mortality in the UNECE region over the studied time window is mainly associated with the growing share of $CH_4$ emissions from ROW. This is mostly related to the relatively long atmospheric lifetime of $O_3$ (compared to $PM_{2.5}$), formed from its short-lived precursors $NO_x$ and NMVOC, and to the one of its long-lived precursor $CH_4$ which contributes to global background $O_3$. On the contrary, $PM_{2.5}$ related mortality in UNECE appears to be mainly affected by its own emissions.

Controlling $O_3$ exposure in UNECE counties is necessary to prevent the CLE projected increase in annual mortality from ca. 65,000 in 2020 to ca. 73,500 in 2050 (+9,000 deaths/year), while acting on $PM_{2.5}$ is a high priority to avoid the considerable mortality attributed to this pollutant turning back in 2050 to the same levels of 2020 (ca. 444,000 units). The analysis of the CLE scenario suggests the opportunity to act on $CH_4$ sources agriculture, energy production, gas flaring and waste management beyond the UNECE region (ROW) in order to prevent an increase in $O_3$ exposure and related mortality in the UNECE countries from 2030 onwards (in addition to the benefits for the ROW region). On the contrary, to significantly reduce the $PM_{2.5}$ exposure and related mortality in the UNECE region beyond the CLE measures in the long term (2050), the main focus should be on the anthropogenic emissions from agriculture and industry sectors within the UNECE region.

In MFR-SDS, the abatement of some of the most critical $CH_4$ sources identified in the analysis of CLE (energy production, gas flaring and waste management) plus the reduction of $NO_X$-VOC from industry and transport globally and those of the maritime sector lead to a 30% - 41% drop of $O_3$-related mortality with respect to CLE in 2030 and 2050 (equal to ca. 20,000 – 30,000 avoided premature deaths/year), respectively. Moreover, the abatement of the most critical UNECE $PM_{2.5}$ emissions identified in the analysis of CLE (i.e. agriculture and industry) plus emissions in the domestic sector complemented by

reductions in natural sources (DUST and SS) lead to a 44% - 50% drop in the $PM_{2.5}$ related mortality compared to CLE in 2030 and 2050 (equal to ca. 182,000 – 221,000 avoided premature deaths/year), respectively.

The analysis of MFR-SDS scenario confirms that the measures in line with UN SDGs concerning energy sources can lead to significant benefits. It also shows the potential co-benefits of joint air quality and GHG abatement policies in line with Paris Agreement ambition of keeping the global average temperature increase below 2°C. However, considering the impact of

agriculture, an important $NH_3$ contributor, on the two studied pollutants in the CLE scenario, more ambitious reductions of this source should be explored considering that the abatement of $NH_3$ in the MFR scenarios compared to CLE is modest (-32 % to -35% in UNECE in the studied time window).

The conclusions of this study are relevant for the revision of the UNECE's Air Convention Gothenburg protocol.

### Code/Data availability

The FASST code is available at https://github.com/JGCRI/rfasst

Data used in this work are available at https://doi.org/10.5281/zenodo.8077436

### Author contribution

Both authors contributed equally to all the phases of the work and manuscript drafting.

### Competing interests

The authors declare they have no conflict of interest.

### Acknowledgements

The authors are grateful to Julian Wilson for reviewing the English style of the manuscript. Special thanks to Zig Klimont and Chris Heyes for sharing an early version of the ECLIPSE scenarios.

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

## Appendix A

**Comparison with other studies**

The source allocation of average PM$_{2.5}$ exposure in UNECE described in the present study is comparable with the one reported by Mc Duffie et al. (2021) for all world countries in 2017 on the basis of a combination of satellite data, chemical transport models and ground based observations. The UNECE average population weighted PM$_{2.5}$ split in 20 source categories including fuel details obtained from the country averages reported in the abovementioned study is shown in Figure A1 left. Such

categories are merged using the same categories as the present study for comparison with the estimations obtained with TM5-FASST extrapolated for 2017 (Figure A1 right).

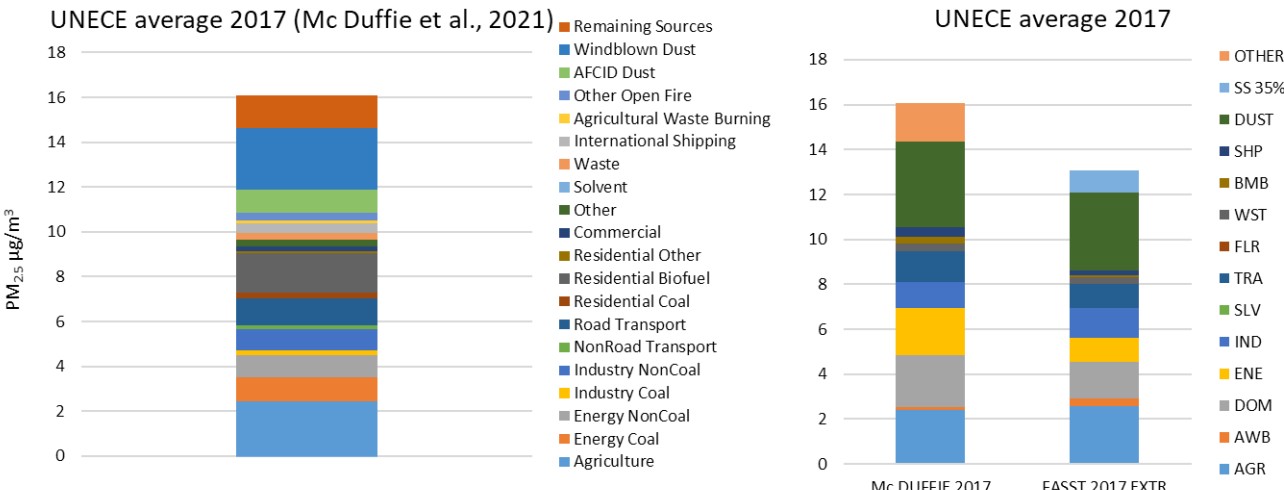

**Figure A1. UNECE average population weighted PM$_{2.5}$ split by source categories. Left: Original source categories (Mc Duffie et al., 2021); Right: comparison of PM2.5 source apportionment of the present study with the one by Mc Duffie et al. 2021 using the same**
**source categories.**

The average UNECE population weighted PM$_{2.5}$ from TM5-FASST is 2.4 μg/m$^3$ (-18%) lower than the one obtained from the country values reported by Mc Duffie and co-authors, likely due to the use of data fusion in the latter. The population weighted PM$_{2.5}$ allocated by TM5-FASST to energy production and domestic is lower than the one reported in the abovementioned study (-47% and -29%, respectively). On the contrary, the higher agricultural waste burning share in TM5-FASST (+160%)

has been attributed to the incorporation of forest fires under this category in this model (Figure A1 right).

The UNECE O$_3$ source allocation in the 2010 warm season (April-September) obtained in this study with TM5-FASST based on a perturbation approach was compared with the one reported by Butler et al. (2020) using a tagging approach (hereon Butler2020). Comparing the O3 apportionment in these studies is, however, not straightforward because Butler2020 splits the total O$_3$ concentrations in two alternative ways either by NO$_X$ precursors or by VOC precursors while TM5-FASST splits them

between NO$_X$–VOC and CH$_4$ precursors at once. Moreover, in Butler2020 Central Asia (CAS) VOC contributions as well as those from Israel are included in ROW while in this study these countries have been included the UNECE region.

The O3 concentrations are higher in TM5-FASST compared to Butler2020 likely due to the use of maximum daily 8h averages instead of monthly averages (Figure A2). The share of $O_3$ produced by $NO_X$-VOC emitted in UNECE according to TM5-FASST (6 ppb, 13%) lies in-between the estimations obtained by Butler2020 for the contribution of $NO_X$ (17 ppb, 45%) and NMVOC (4 ppb, 10%) emissions in this region. By comparison, the share of $O_3$ deriving from $NO_X$-VOC emissions from ROW provided by TM5-FASST (2 ppb, 4%) is slightly lower than the estimations by Butler2020 for $NO_X$ (4 ppb, 11 %) and VOC (3 ppb, 7%), respectively.

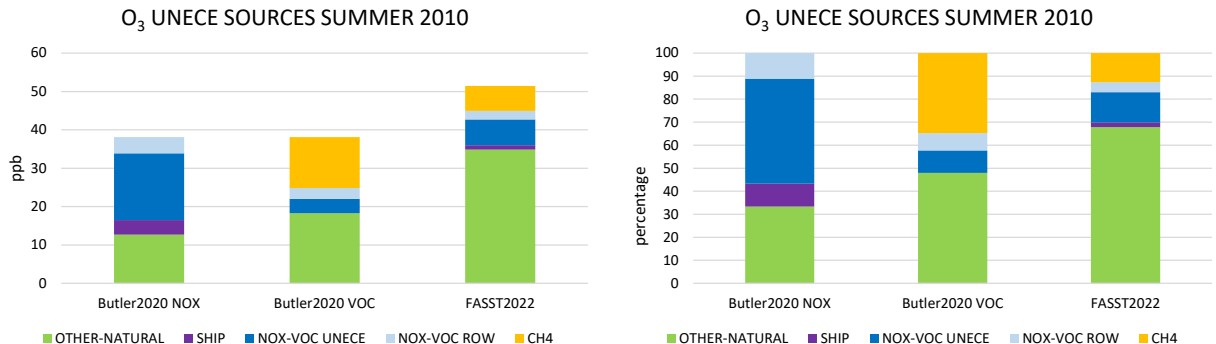

**Figure A2. UNECE average O3 split by sources categories using a tagged approach (Butler et al., 2020) and a perturbation approach (TM5-FASST, this study) expressed as concentrations (left) and percentages (right).**

Butler2020 links the $CH_4$-related $O_3$ only to VOC emissions and does not associate this precursor to any specific geographic area while TM5-FASST allocates $CH_4$-related $O_3$ to its geographic source regions and precursors. In this analysis the TM5-FASST aggregated share of $O_3$ associated with $CH_4$ (6 ppb, 13%) is considerably lower than the one attributed by Butler2020 to this fraction (13 ppb, 35%). Also the contribution of shipping to $O_3$ concentrations estimated by Butler2020 (4 ppb, 10%) is higher than the share reported by TM5-FASST in this study (1 ppb, 2%). By comparison, the role of Other-Natural source is higher in TM5-FASST (35 ppb, 67%) compared with the one attributed by Butler2020 (13 ppb, 33% for $NO_X$ and 18 ppb, 48% for VOC source allocation, respectively).

### Brief description of scenarios

The scenarios used in this study are summarised in Table A1.

**Table A1. Description of ECLIPSE version 6b global scenarios used in this study (IIASA, 2021).**

| Scenario | abbreviation | Air quality policy | Climate policy |
|---|---|---|---|
| Current legislation (baseline) | CLE | Assumes the implementation of the future commitments included in the air quality legislation in force by 2018. Current baseline | Incorporates only commitments made in the national determined contributions (NDC) under the Paris Agreement. |

| | | | |
|---|---|---|---|
| | | projections according to the IEA World Energy Outlook 2018 New Policy Scenario (NPS) which includes EU 2030 renewable energy and energy efficiency targets and announced energy policies by China, USA, Japan and Korea. | |
| Maximum technical reduction baseline | MFR BASE | Stringent policy assuming introduction of best currently available technology and no cost limitations. However, no further technological improvements are foreseen. Same activity drivers as CLE following NPS. | Incorporates only commitments made in the NDCs under the Paris Agreement. |
| Maximum technical reduction sustainable development | MFR-SDS | Similar to MFR BASE. However, relies on the most ambitious IEA sustainable development scenario (SDS). Includes outcomes of energy-related SDGs: reducing dramatically premature deaths due to energy-related air pollution and universal access to modern energy by 2030. | Aligned with Sustainable Development Goal #13 and Paris Agreement goal of holding global average temperature increase below 2 °C. |

The current legislation baseline (CLE) scenario considers fuel consumption from IEA (International Energy Agency), agriculture data from FAO (UN Food and Agriculture Organisation) and IFA (International Fertilizer Organization), and statistics on industry, waste, shipping, etc., from other sources (IEA, 2018).

535