# Peer review of "Air quality and related health impact in the UNECE region: source attribution and scenario analysis"

_Atmospheric Chemistry and Physics, 2022_

## Author Comment (AC1)

**Answers to reviewers (Manuscript ACP-2022-776)**

**Anonymous Referee #1**

This is an interesting paper addressing the influence of in- vs. out-of-region sources on air quality and health impacts in the UNECE region. It uses a well-documented tool, and provides scenarios that will be of interest to policy-making. Overall, it is a strong analysis and in my view could be published with minor revisions.
There are two general areas in which the manuscript could be improved to more effectively communicate its results.

First, the scenarios could be better described in the paper itself, including the selection of scenarios and the distinctions between the air quality focused and climate focused policies included in them. While these are presented briefly (and the table is helpful) more discussion in the manuscript itself (rather than the cited references) on why these scenarios were chosen and what policy outcomes they are meant to address would be useful.

Answer
Section 2.3 has been expanded to describe how the scenarios have been developed and examples of key measures have been included in the description. The motivation for the choice of scenarios is now mentioned at the end of the introduction (Section 1).

Second, the limitations of the model as a reduced-form approach could be clearer. The current paragraph towards the end of the manuscript is a good start, and I understand from the manuscript that nonlinearities in chemical regimes aren't included. However, I don't have a good sense from reading the manuscript of when/where the model is most trustworthy, and what should be interpreted with caution, beyond this general statement.

Answer
In the discussion section we clarify that the TM5-FASST satisfactorily reproduces a full chemical transport model using source receptor coefficients obtained with the meteorology of a specific reference year. However, modelling the feedbacks associated with changing climatic conditions is beyond the capabilities of the model.

Some attention to details here might help the reader better understand the tool itself. In particular, some description of how methane is treated would be particularly useful (as it's not obvious in a source-receptor framework what timescales are examined and how they are considered).

Answer
The CH4 – ozone interactions in the TM5-FASST model are described in the supplementary material and in section 2.1. In addition, the implications of the model assumptions concerning this precursor are now discussed in section 4 of the manuscript.

Minor comments:

TM5-FASST model description: it would be useful to include some of the key description of the model in the paper itself, as well as including more detail in the supplementary information, such that the paper is

able to stand alone. For example, what form do the source-receptor coefficients take -- is this a linear pollutant-by-pollutant approach? How are combinations addressed? How is the downscaling done?

Answer
The model source-receptor coefficients, linearization and downscaling approaches are now described in the supplementary material

Line 65-75, and later discussion: the acronyms here make the paper difficult to read and interpret, especially in the discussion in section 3. (Line 178-180 is a key example, where it reads "The contributing sectors change their relative importance evolving from a mix dominated by TRA, AGR and ENE in 2020 to a one dominated by AGR, WST, TRA and ENE in 2050.") Perhaps consider using more obvious names for these sources?

Answer
The extended names of the sources are now used in the text instead of the acronyms

Figure 2: it is difficult to compare the totals directly across the different scenarios; might this figure be combined/streamlined?

Answer
The figure has been restructured for a straightforward visualization and comparison of the emissions in the different scenarios.

Section 3.2 -- is the regional breakdown only done for the MFR scenarios? It might be interesting to look at and compare whether degrees of reduction policy affect the results presented in Figure 3 (at least that wasn't completely clear from the discussion as far as I can tell).

Answer
The objective of section 3.2 is to provide evidence about the limitations of implementing abatement policies only in the UNECE region to achieve significant reduction of pollution levels in this region. The analysis of different degrees of reduction, represented by the two MFR scenarios, with a break down by region, precursors and anthropogenic sources is presented in section 3.5

Similarly, it would be useful to have a comprehensive list of all of the simulations/sectors/combinations (perhaps a modified version of Figure 1 to be more overarching?)

Answer
A new paragraph was added in section 2.2 describing more in detail all the simulations and how they have been combined to obtain information about sources, geographic origin and precursors.

Figure 4 and similar bar charts -- it would be good to include the numbers in a supplementary table. The pie charts are also hard to read -- providing numbers, or using bar charts, might be helpful. Some of the figures could also be cut where the main messages are just as easily communicated in text, or moved to supplementary information.

Answer

The pie charts in the old Figure 4 were converted into a bar chart and merged with the old Figure 5 in a single Figure (new Figure 4). Similarly, the old Figure 7 was converted into a bar chart and merged with old Figure 8 to create the new Figure 6.
In addition, two new tables were added in the supplementary material with the numeric information included in old figures 6 and 9 (now figures 5 and 7)

Line 309-310 makes a point about the potential co-benefits of joint air quality and GHG abatement policies, which is important, but was a bit lost in the details earlier. I see this only when reviewing Table A1, which specifies which policies are addressed under which scenarios, but I'm not 100% sure even after re-reading whether the CH4 policies are only addressed under "climate" or whether there are others relating to "air quality"? Some clarification, or potentially a short description of the types of policies considered under the scenarios, could be useful for a broader practitioner audience that is not familiar with the specific scenarios used.
Answer
To our understanding in the Eclipse v6b scenarios the CH4 abatement is mainly associated with climate policies as in the past the role of this precursor on air pollution was not well known. In the revised section 2.3 we have included examples of cross-cutting measures and measures for specific regions in the different scenarios. However, a detailed description of the scenarios is much beyond the scope of the present work and, therefore, the reader is redirected to the original literature sources.

**Anonymous Referee #2**

The authors use the TM5-FASST reduced-order air quality model to evaluate how air quality in the UNECE region might vary between 2020 and 2050. Specifically, they investigate the relative contribution of UNECE and non-UNECE emissions to UNECE air quality degradation, and how this varies under different future emissions scenarios. The study finds that enacting ambitious air quality legislation would likely significantly reduce mortality in the UNECE region, and that the dominant contribution to this improvement is from reductions in UNECE emissions. The study also shows that there would likely be an additional air quality cobenefit of implementing ambitious climate policy.
The central question of the study is interesting and important. The TM5-FASST model is an appropriate tool for investigations of air quality impacts from emissions changes and the scenarios investigated are relevant. However, it is not clear to me that the TM5-FASST model can accurately represent the interaction of climate change, reductions in NOx/VOC emissions, and changes in methane emissions. Nonetheless, I believe that – if the authors are able to address the concerns listed below – the paper is in principle publishable.

Major comments
One conclusion of the study is that CH4 becomes more important, while NOx and NMVOCs become less important. However, can TM5-FASST capture the interactions between an increased level of CH4 and the total ozone production per unit of NOx emitted?
Answer
The sensitiveness of the TM5-FASST model to changes in CH4 concentrations is described in the supplementary material and in section 2.1. Moreover, the influence of the model assumptions on the results are now discussed in section 4 of the manuscript.

TM5-FASST's approach to calculating the impacts of methane emissions is an estimate derived from simulation results for a global change in methane concentration (van Dingenen et al., 2018). Given the prominence of methane changes in the authors' conclusions, a quantitative evaluation of the degree to which TM5-FASST can or cannot accurately represent the impacts of combined changes in NOx/NMVOCs and methane would significantly improve the paper. I would recommend that the authors consider doing so if possible.

Answer
To answer this question we have added the following paragraph in section 4: "The impact of increased natural VOC (including methane) on the O3 response to NOx emission changes depends on the chemical regime. In NOx-saturated (VOC-limited) conditions, the climate-driven increased VOC emissions will increase the natural component of O3 formation, and drive the chemical regime more towards the NOx-limited region, implying a higher response of O3 to anthropogenic NOx emission changes. This situation is only characteristic for strongly polluted urban areas. Under the more common conditions of VOC-saturation (NOx-limitation), the O3 response to NOx is only weakly dependent on the VOC concentrations (Akimoto and Tanimoto, 2022 )."

Similarly, it appears that the effects of climate change on emissions sensitivity are not only not included, but not discussed. This seems like an oversight. As a straight-forward example, von Schneidemesser et al. (2020) discuss the need for holistic assessment of climate and air quality policy, and both Jacob et al. (2009) and Fiore et al. (2015) discuss the possibility that the sensitivity of air quality to pollutant emissions could be affected both positively and negatively by climate change. I recommend that the authors at minimum include a summary discussion of how the lack of representation of the effects of climate change may affect their conclusions.

Answer
The present study focuses on long-range transport of anthropogenic emissions. Climate-chemistry feedbacks are beyond the scope and capacities of the TM5-FASST model. This is now explained in the revised section 4 of the manuscript.

Minor comments
The method used appears to treat impacts in UNECE as responding uniformly. This may be an inherent limitation of TM5-FASST, but it would be helpful to consider how the impacts of the different interventions might be differently distributed within the UNECE region, and if that distribution might be expected to change with future population movement or with future changes in background emissions.

Answer
The TM5-FASST tool used for this study includes 56 pollutant emission sources 26 of which are single UNECE or small group of UNECE countries. Moreover, the output of the tool is at country level and covers all the UNECE States with the exception of very small ones (Lichtentstein, Monaco, etc.). So we are able to provide detailed analysis of the geographical gradients within the region. We have now expanded section 3.2 to identify the UNECE countries where the influence of non-UNECE emissions is the highest. However, considering the complexity of the study which splits the results of three scenarios by main geographical regions, anthropogenic sources and ozone precursors, we believe that going further into the internal variability among UNECE countries would be detrimental for the extension and readability of the manuscript.

It is difficult to extract a clear and convincing message, especially regarding the claim of an air quality-climate policy co-benefit (lines 309-310). This seems to be more related to the form of the discussion rather than an actual lack of content. If the authors do want to make this statement, then (for example) the difference between MFR-SDS and MFR-BASE could be communicated as the "climate policy-driven" reduction in mortality, compared to the "air quality policy-driven" reduction when comparing CLE to MFR-BASE.

Answer
Actually both scenarios are a combination of air quality and climate-oriented policies. So we prefer to present them as different levels of ambition. The MRF BASE relying most on air quality abatement technologies and a basic set of climate measures (NDC) and the MFR-SDS mostly driven by ambitious measures in the energy sector in line with UN sustainable development goals (e.g. $CO_2$ emission pricing, phasing out fossil fuel subsidies, imposing maximum sulfur content in fuels, etc.) and climate measures to achieve 2 °C global temperature increase target. We have now included a more detailed description of scenarios in section 2.3. However, a full description of the scenarios goes far beyond the purposes of the present work. The reader is oriented to the relevant literature.

Part of the difficulty here is because of the heavy reliance on acronyms and abbreviations, so I would recommend that the authors consider trying to present their results in more intuitive ways (e.g. the effects of "agricultural emissions" rather than of AGR).

Answer
The extended names of the sources are now used in the text instead of the acronyms

The aforementioned limitations regarding the representation of the effect of climate change on linear sensitivities would also need to be addressed, or at least discussed.

Answer
The limitations of the adopted approach with respect to future climate changes are discussed in the revised section 4 of the manuscript.

The abstract discusses UNECE but does not define it – please add a definition on first use.

Answer
The list of UNECE member States has been added in a footnote at the first use.

References

Fiore, A. M., Naik, V., and Leibensperger, E. M.: Air quality and climate connections, J. Air Waste Manag. Assoc., 65, 645–685, 2015.
Jacob, D. J. and Winner, D. a.: Effect of climate change on air quality, Atmos. Environ., 43, 51–63, 2009.
Van Dingenen, R., Dentener, F., Crippa, M., Leitao, J., Marmer, E., Rao, S., Solazzo, E., and Valentini, L.: TM5-FASST: a global atmospheric source–receptor model for rapid impact analysis of emission changes on air quality and short-lived climate pollutants, Atmos. Chem. Phys., 18, 16173–16211, 2018.
von Schneidemesser, E., Driscoll, C., Rieder, H. E., and Schiferl, L. D.: How will air quality effects on human health, crops and ecosystems change in the future?, Philos. Trans. A Math. Phys. Eng. Sci., 378, 20190330, 2020.

---

## Author Response (AR2)

**Answers to R1 reviewers' comments (Manuscript ACP-2022-776)**

**Report #1**

Anonymous referee #2

The authors have responded adequately to almost all of my concerns, and I concur that further extensions would dilute the paper without necessarily increasing its impact or utility. I in particular appreciate the new text on limitations which I think goes a long way to helping ensure that the manuscript. However, I do have two remaining issues which I believe should be addressed prior to publication.

Answer

The authors are grateful to the reviewer for the positive comments about version R1 and the useful recommendations.

First, the description of the effect of climate change on air quality (the so-called "climate penalty") is somewhat misleading. The effect of climate on surface ozone is complex, and the implication that it will inevitably result in an increase in surface ozone (lines 337-340) is incorrect. Jacob and Winner (2009) is cited but said study finds that the effect of climate change on ozone is mixed, rather than purely increasing ozone. More recent reviews, such as Fiore et al. (2015), provide elegant explorations of the complex relationship between climate change and ozone. I recommend that the authors revise their description of the effects of climate change on air quality to reflect this nuanced issue more accurately.

Answer

We have revised Section 4 (new lines 339 – 357) to better reflect the complexity (and related uncertainties) of the interaction between climate and air pollution including examples and citing relevant bibliography as suggested by the reviewer.

Finally, the current code and data availability statements are very weak. I would urge the authors to make their code and data publicly available as recommended under the standards of open science (e.g. FAIR principles). This would dramatically improve both the reach and the reproducibility of the work.

Citation: Fiore, A. M., Naik, V., and Leibensperger, E. M.: Air quality and climate connections, J. Air Waste Manag. Assoc., 2015.

Answer

We have now added public links to the FASST model code in Github and the dataset was published in the Zenodo repository. We have also made a small correction to Figure 1 in order to align the terminology with the dataset.

**Report #2**

Anonymous referee #1

**For final publication, the manuscript should be**

accepted as is